# Disease trajectory browser for exploring temporal, population-wide disease progression patterns in 7.2 million Danish patients

Troels Siggaard[1], Roc Reguant [1], Isabella F. Jørgensen[1], Amalie D. Haue [1], Mette Lademann[1], Alejandro Aguayo-Orozco [1], Jessica X. Hjaltelin[1], Anders Boeck Jensen[1,2], Karina Banasik [1] & Søren Brunak [1✉]

We present the Danish Disease Trajectory Browser (DTB), a tool for exploring almost 25 years of data from the Danish National Patient Register. In the dataset comprising 7.2 million patients and 122 million admissions, users can identify diagnosis pairs with statistically significant directionality and combine them to linear disease trajectories. Users can search for one or more disease codes (ICD-10 classification) and explore disease progression patterns via an array of functionalities. For example, a set of linear trajectories can be merged into a disease trajectory network displaying the entire multimorbidity spectrum of a disease in a single connected graph. Using data from the Danish Register for Causes of Death mortality is also included. The tool is disease-agnostic across both rare and common diseases and is showcased by exploring multimorbidity in Down syndrome (ICD-10 code Q90) and hypertension (ICD-10 code I10). Finally, we show how search results can be customized and exported from the browser in a format of choice (i.e. JSON, PNG, JPEG and CSV).

---

[1] Novo Nordisk Foundation Center for Protein Research, Faculty of Health and Medical Sciences, University of Copenhagen, DK-2200 Copenhagen, Denmark. [2] Institute for Next Generation Healthcare, Icahn School of Medicine at Mount Sinai, New York, NY 10029-6574, USA. ✉email: soren.brunak@cpr.ku.dk

L large-scale, population-wide studies of health and disease are increasingly gaining attention, as they complement conventional clinical studies focusing on a limited set of hypothesised disease associations to predefined outcomes[1–4]. However, most population-wide disease registries are made available for research as large databases that require custom-made software to be analysed[5–7]. While some disease registries provide interfaces that can extract summary statistics for patients with a given diagnosis, more advanced queries on disease progression patterns are generally not supported[8]. Multimorbidity—instances where one patient is diagnosed with more than one chronic morbidity—is, as life expectancy generally goes up, an increasing problem[9]. Therefore, the need to understand at the molecular level how etiological factors impact co-occurring and interacting diseases is also growing, where strategies typically benefit from the application of network biology concepts[1,10,11]. However, these efforts should ultimately match disease progression observations made in large-scale, population-wide health data as already demonstrated in uncovering associations between complex disease and Mendelian loci[2] and estimates of heritability in the absence of genetic data[4].

In clinical trials, the complexity of multimorbidity is being addressed by master protocols defined as an overarching model designed to answer multiple questions, for example, by studying multiple interventions in multiple diseases[12]. Correspondingly, adaptive platform trials (APTs) facilitate the study of multiple interventions in a condition in a perpetual manner. In these trials, interventions may enter and leave an ongoing trial based on a predefined decision algorithm, e.g., response-adaptive randomisation (RAR)[13]. The shift towards more complex trial designs again emphasises the demand to monitor and study phenotypes as longitudinal disease progression patterns irrespective of predefined outcomes[3,14]. At the molecular level, techniques like whole-genome sequencing and clinical proteomics produce data in a disease spectrum-wide manner enabling analysis of global interactions between disease pathways for subsequent linkage to actionable mechanisms[15–17]. Finally, another contributing factor is the health economic incentive to study multimorbidity as medical management is complex and costly[18].

The aim in numerous ongoing precision medicine efforts is to stratify seemingly similar patients into subgroups by identification of underlying differences in disease etiology[1,19]. Ultimately, this will reduce the gap between geno- and phenotypes for optimised diagnostics and enhanced therapeutic responses[20]. In line with the precision medicine discourse, disease trajectories, derived from population-wide registry data, were recently proposed as a novel robust way of studying disease progression over time; thereby creating a foundation for the discovery of causal relationships and the temporal modelling of multimorbidity[3,21,22]. Trajectories illustrating frequent and recurrent patterns of disease progression can be constructed by identifying pairs of co-occurring sequential diseases with a statistically significant direction[23]. Such work is likely to lead to revisions in the disease nomenclature; for example by redefining diseases based on disease history data and genetic risk profiles as opposed to the present paradigm where conclusions are derived from single encounters with the health care system[24,25].

In this context, we have constructed The Danish Disease Trajectory Browser (DTB) that lets users, e.g. researchers, trialists and clinicians explore longitudinal population-wide disease progression patterns from the entire population of Denmark. The browser is based on data spanning the period January 1994 to April 2018 and comprises electronic health data on 7.2 million Danes representing a total of 122 million admissions during the almost 25-year long period. Denmark implemented its unique person identification number in 1968 making it possible to track

disease development continuously across all hospitals at the level of individuals[26,27]. DTB enables users to search, filter, analyse, and visualise disease trajectories derived from, statistically significant directional diagnosis pairs calculated from population-wide electronic health data[3,21]. As the underlying data stem from a government-funded universal health care system, the data are likely less biased than many other large data sets that either focus on selected diseases, age groups, hospitals, or professions[6,27]. DTB is made available at http://dtb.cpr.ku.dk and presents data as summary statistics. Therefore, it does not provide access to any person-sensitive data.

## Results

**Data foundation and disease trajectory approach.** The data foundation for DTB is the Danish National Patient Register (NPR) covering 7.2 million patients from the 1994–2018 period. All entries in NPR are time stamped and linked to the national identification number that uniquely indexes every resident in Denmark. The sex is indexed in the final digit (i.e. unequal, male) [27]. Events are therefore linked over time unproblematically, which this study benefitted from in computing mortality by linkage of patients from NPR to The Danish Register of Causes of Death[28]. DTB uses a general approach to create disease trajectories that was originally developed to discover sequential disease progression patterns in a data-driven manner[3,14]. Following a pre-filtering step and a matching procedure that corrects for age, sex, and period of year, all 1777 unique disease codes (International Statistical Classification of Diseases and Related Health Problems (ICD) version 10, level 3 indexed) resulted in 77,294 significant diagnosis pairs (Fig. 1). For subsequent analyses of disease progression patterns, the method requires that the direction of diagnoses D1 -> D2, is statistically significant compared to the reverse order D2 -> D1 with relative risk (RR) > 1 and P-value < 0.05 (Fig. 1). Disease trajectories must be followed by minimum 20 patients. The directional progression strength D1 -> D2 can then be quantified by the RR and the associated P-value for any significant directional disease pair that results from a search query[21] (see Methods). Furthermore, the DTB offers the option to analyse sex-specific disease trajectories, which are built from significant directional diagnosis pairs computed from male or female populations only. In effect, DTB users can also explore differences in disease progression patterns between men and women.

**Disease progression patterns and disease heterogeneity.** From the complete data set a total of 9608 statistically significant directional diagnosis pairs were identified (Table 1 and Fig. 1). These diagnosis pairs form the basis for the trajectory browser functionalities as they can be combined into linear trajectories and disease trajectory networks as shown in previous studies[3,14]. The disease trajectory D1 -> D3 cannot necessarily be deduced from the combination of D1 -> D2 -> D3. This highlights the complexity of disease heterogeneity and disease progression patterns. DTB queries for one or more ICD-10 codes return a number of trajectories with length ranging from 2 to 6 depending on the search base entry (ICD-10 codes) and search filter selection.

Below we exemplify the browser functionalities and potential for exploring population-wide disease progression patterns by characterising search results for patients diagnosed with Down syndrome (DS) and patients diagnosed with essential (primary) hypertension (HT; ICD-10 codes Q90 and I10, respectively), which represent two very different cohorts. We also exemplify how the combination of length 2 trajectories into longer trajectories further elucidates disease heterogeneity. The browser

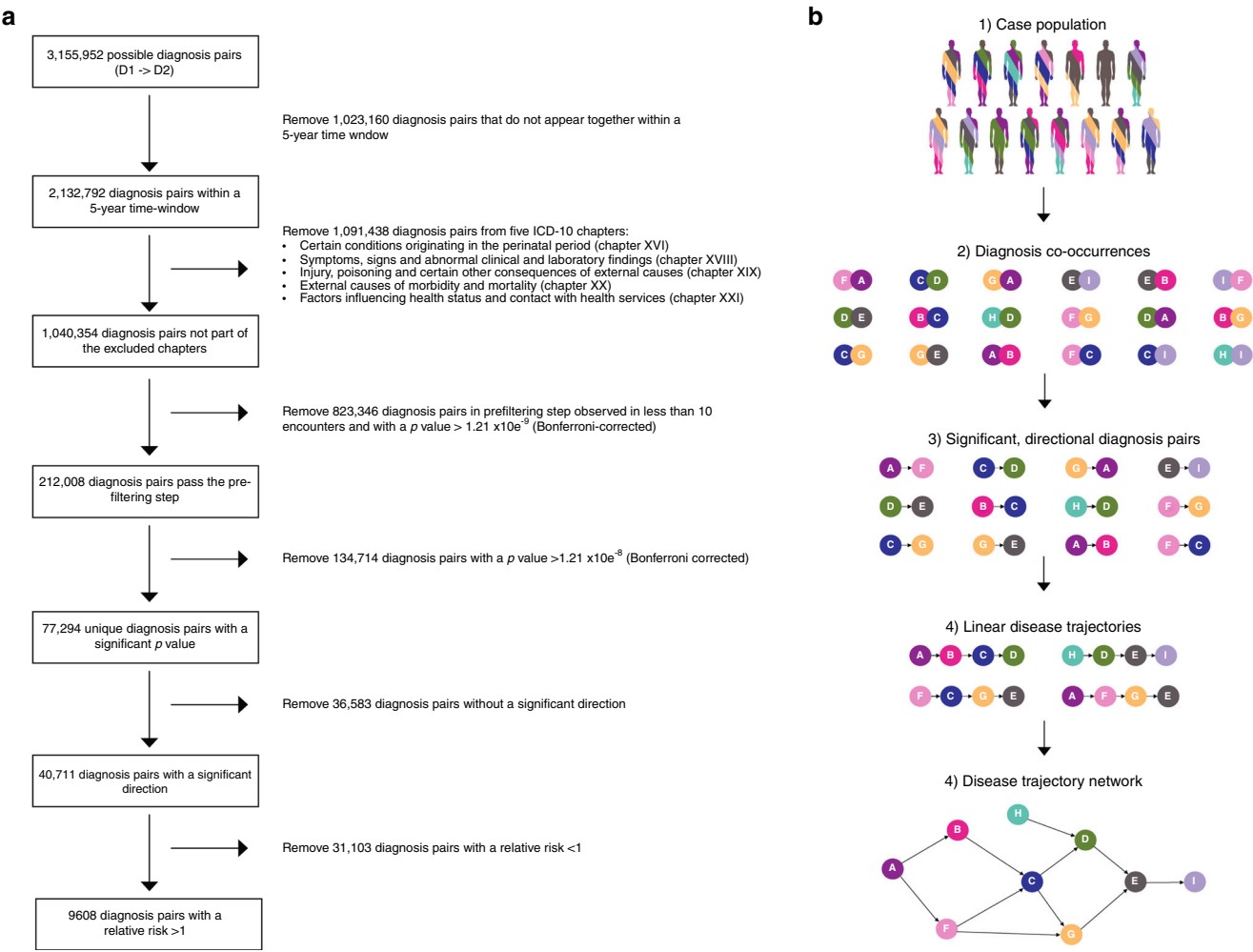

**Fig. 1 Register data processing.** Disease trajectory algorithm flowchart (**a**) and schematic representation of the disease trajectory algorithm that is the foundation for Danish Disease Trajectory Browser functionality (**b**). **a** Pre-filtering and computation of statistically significant diagnosis pairs in The Danish National Patient Registry (NPR), containing data on 7,186,865 patients. **b** (1) Users select a ICD-10 indexed patient population by typing ICD-10 code(s) or disease(s) of interest (ICD-10 code, level 3). (2/3) The first step where the algorithm identifies statistically significant directional diagnosis pairs. (4/5) The second step where the algorithm builds linear trajectories by concatenating statistically significant diagnosis pairs, that can then be merged into a disease trajectory network, consequent to the fact that one disease may appear in more than one linear disease trajectory. ICD-10 International Classification of Diseases, 10th Revision. Colours indicate disease category.

**Table 1 The Danish Disease Trajectory Browser data foundation and examples.**

| The Danish National Patient Registry | | | |
|---|---|---|---|
| Years of data | 24 | Total number of hospital admissions | 121,884,394 |
| Total number of patients | 7,186,865 | Number of unique ICD-10 codes given[a] | 1777 |
| The Danish Disease Trajectory Browser | | | |
| Total number of directional diagnosis pairs | 9608 | Number of ICD codes included in the browser in directional pairs[a] | 928 |
| Down syndrome (ICD-10 code: Q90) | | | |
| Number of patients | 3714 | | |
| Number of linear disease trajectories | 117 | | |
| Essential (primary) hypertension (ICD-10 code: I10) | | | |
| Number of patients | 807,234 | | |
| Number of linear disease trajectories | 8359 | | |

*ICD* International Statistical Classification of Diseases and Related Health Problems.

displays search results either as linear disease trajectories or as a disease trajectory network (e.g. Figs. 2 and 3). For quick queries and overview, it is recommended that users turn on the network view, which will be fastest and also appropriate for subsequent adjustment of search filters. In DS (n = 3714 patients) and HT (n = 807,234 patients), the number of resulting linear trajectories ranges from 117 to 8359 (Table 1). These disease codes relate to a chromosomal disorder and multifactorial disease, respectively, and hence, the two cohorts have vastly different frequencies and only limited shared etiology. Consequently, the number of

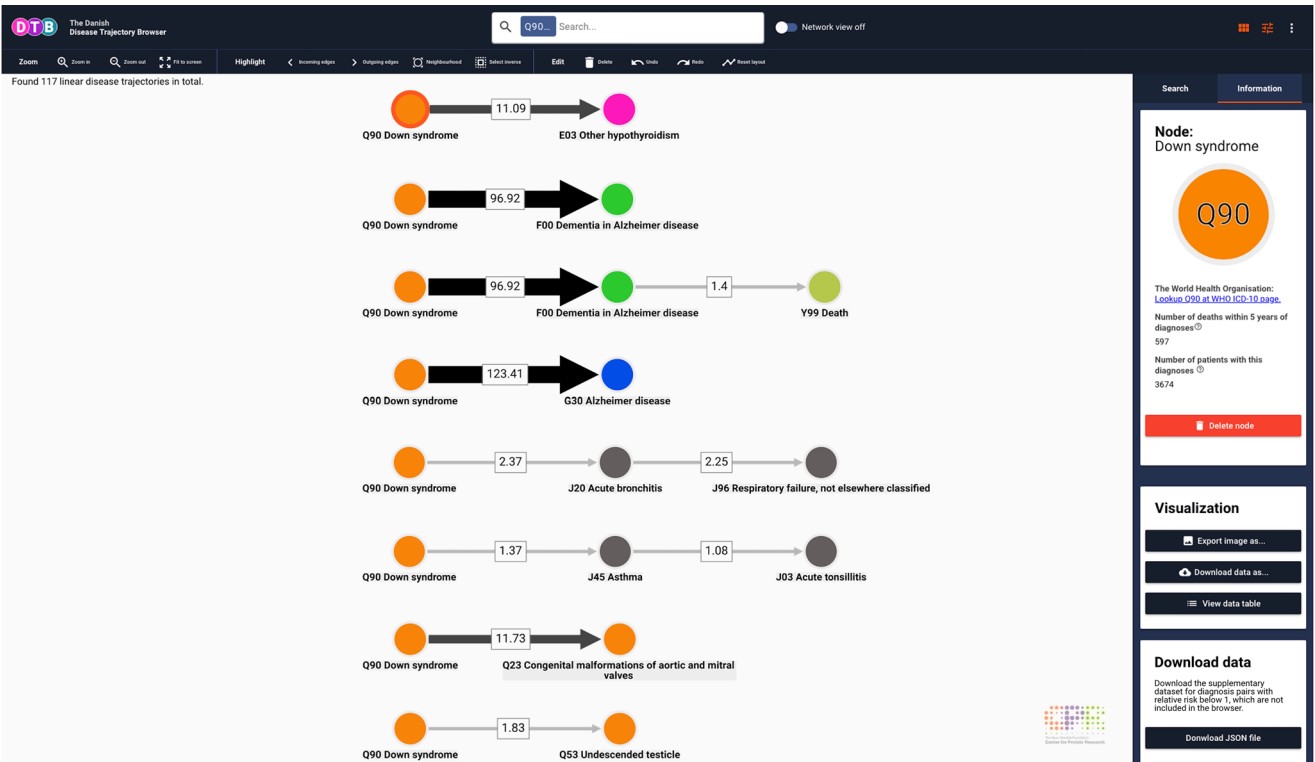

**Fig. 2 Linear disease trajectories for Down syndrome (Q90).** Upper bar with search field and switch to turn on Network view. Below the upper bar search results are displayed in the visualisation interface. Users can see the total number of linear trajectories (in this case 117). When a node is selected, search results are linked to the WHO ICD-10 browser in the side panel. The content of the link depends on the selected node (circles). When an edge is selected, search results also link to Google Scholar and Statistics Denmark. Width and shade of edges co-vary with the number of patients that follow the particular trajectory or as here the relative risk (depending on Settings selection, located in the side panel, Search tab). Users can navigate the search results, select the content of the links and change location of trajectories by clicking and dragging in the search result field. Search filter setting: no settings applied. WHO ICD-10 World Health Organization, International Classification of Diseases, 10th Revision. Colours of nodes represent ICD-10 chapters.

resulting trajectories and degree of diversity may hint DTB users towards the underlying complexity of the true multimorbidity spectrum that necessitates protocols and strategies for studying multiple diseases and multiple therapies jointly (e.g. by master protocols as mentioned above)[12].

**Exploring disease trajectories of chromosomal diseases.** As expected, the linear disease trajectories for DS shown in Fig. 2 contain multiple diagnoses from chapter X (Diseases of the respiratory system) and chapter XVII (Congenital malformations, deformations, and chromosomal abnormalities), whereas the disease trajectories contain no diagnosis from chapter II (Neoplasms)[29,30]. These findings may very well relate to the phenotypic effect of gene copy-number, which is rather profound in a trisomy like DS[31]. Moreover, the disease trajectory network topology shown in Fig. 3 reflects the fact that DS is diagnosed early in life (i.e. pre- or perinatally), as Q90 is the first code in all trajectories. It also demonstrates the complexity of disease progression patterns, as there is only a significant directional association among DS patients who are diagnosed with H65 (Nonsuppurative otitis media) and then F79 (Unspecified mental retardation) and no direct edge (i.e. length 2 trajectory) from Q90 to F79 (Fig. 3). Importantly, there is a considerable overlap between diagnoses that people with DS have a high risk for developing, e.g., Alzheimer's disease (ICD-10 code G30)[32] and also diagnoses likely to be listed in the death certificate of people with DS, and the ICD-10 codes present in the trajectory network. These include Other hypothyroidism (ICD-10 code E03) and Congenital malformations of aortic and mitral valves (ICD-10

code Q23)[33,34] (Fig. 2). Further, note the statistical significant pair from Dementia in Alzheimer's disease (ICD-10 code F00) to Death (introduced with code Y99, see Methods section).

**Exploring disease trajectories of multifactorial diseases.** Due to the marked demographic differences between phenotypes and degree of disease heterogeneity, appropriate search filters vary across DTB queries. For instance, proper use of the search filter functionalities is essential in collapsing the 8359 linear disease trajectories originating from an I10 search to a disease trajectory network. For demonstration purposes, we applied search filters for HT resulting in 2672 linear trajectories that were then collapsed into a disease trajectory network (Figs. 4 and 5).

In contrast to the network for DS, the network for HT mainly includes other, multifactorial, chronic diseases, e.g. heart failure (ICD-10 code I50), and other anaemias (ICD-10 code D64); chronic and unspecified kidney failure (ICD-10 codes N18 and N19; Fig. 6). The network topology also reflects this observation as I10 has both incoming and outgoing neighbourhood edges, collectively termed transition edges (Fig. 5). This finding is evidence that HT inherently is a heterogeneous phenotype that is associated with heterogenous progression patterns[35–37] agreeing with the fact that target pressure depends on comorbidities and that comorbidities may even be the treatment indication, per se[38,39]. In addition, the relatively large proportion of linear trajectories length >3, indicates that the average age in this population is rather old and multimorbid. This is further evidence that the I10 population represents a health economic burden as there is a directional, statistically significant association

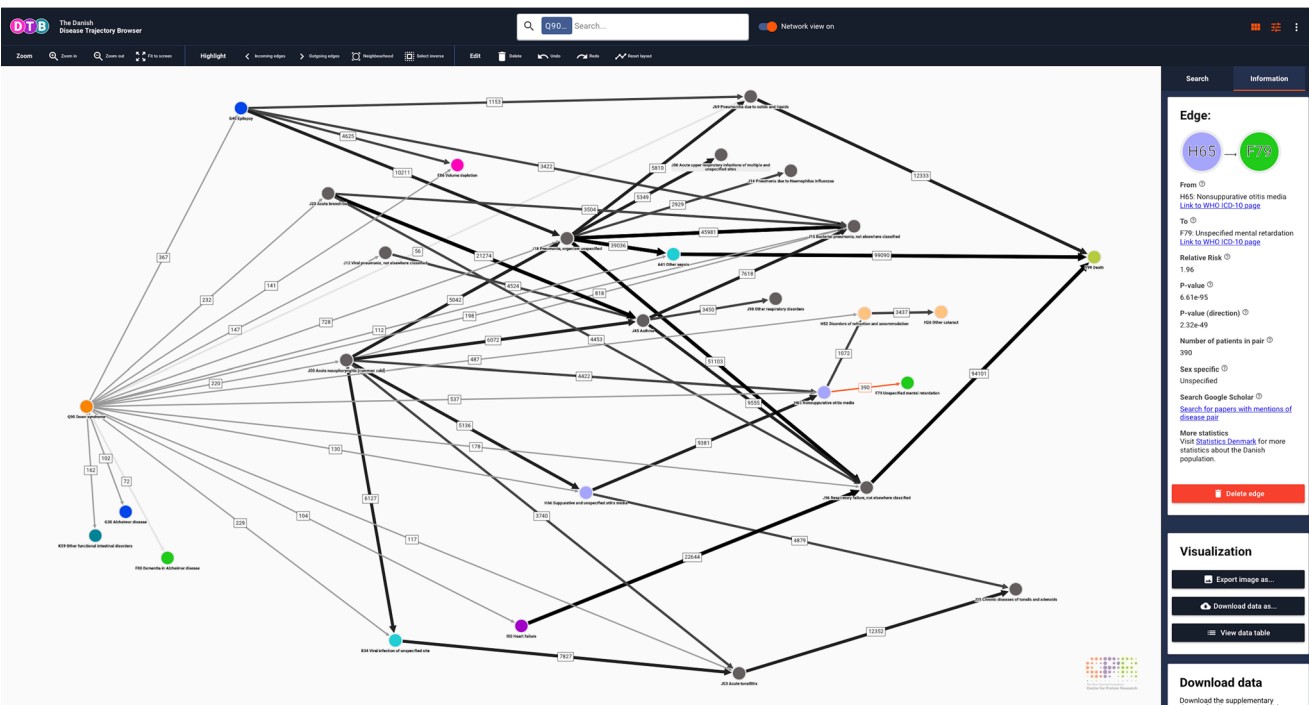

**Fig. 3 Network view representation of the linear disease trajectories for Down Syndrome (Q90).** Nodes represent ICD-10 codes and each colour corresponds to one of the 21 chapters in the ICD-10 index. Five chapters are excluded (for details, see Methods section). Arrows represent a statistically significant directionality between two diagnosis codes. Numbers describe how many patients follow the trajectories. If an edge is selected manually by clicking (red highlight) the statistical specifications will be displayed in the side panel. In the visualisation interface, selected nodes and edges may also be deleted (grouped or individually) by clicking on the icon "Delete edge". A description of each variable can be retrieved by placing the curser over the question marks. Search filter settings: all patients, length 3–5, relative risk (RR) 1.5–361, number of patients 21–114,099. ICD-10 International Classification of Diseases, 10th Revision. Colours of nodes represent ICD-10 chapters.

between e.g. hypertension and heart failure, a rapidly growing public health issue[18,40] (Fig. 6). Interestingly, no malignant disease (indexed in ICD-10 chapter II, Neoplasms) appear in the search on hypertension. This seems to agree with the conflicting literature on the association between cancer risk and hypertension[41] but might as well relate back to a rather pragmatic index structure of the ICD-10 system[42], as opposed to a network-based approach to human disease[11].

**Browser search functionality and trajectory manipulation**. DTB is disease-agnostic in the sense that any level 3 ICD-10 code or combination of ICD-10 codes (for details, see Methods section) can be selected as a search query and then interrogated for involvement in statistically significant disease trajectories.

DTB search results can be visualised in two formats depending on the Network view status (e.g. Figs. 2 and 3). When Network view is off, the browser will list the search result as linear disease trajectories and also report the total number of linear disease trajectories that matches the query (Figs. 2 and 4). When Network view is on, the search result will be displayed in a summarised, collapsed format. Every disease trajectory can be explored further by clicking on nodes (circles) or edges (arrows) that link to the WHO ICD-10 browser, Google Scholar and the Danish National Statistics Bureau in the browser side panel (Fig. 3). When a single node is selected the number of unique patients who are assigned the code within the registry period as well as the 5-year mortality are shown (Fig. 2). When an edge is selected, two different sets of summary statistics appear in the Information tab, depending on the network view selection (Figs. 3 and 4). If network view is on (e.g. Fig. 3), summary statistics for the length 2 disease trajectories, i.e., mean age at diagnosis of D1, time between

diagnoses and 5-year mortality are presented in the side panel. If network view is off users can also view summary statistics for length 2 disease trajectories and the entire disease trajectory of length up to 6 (Fig. 4).

**Search-dependent data filtering and exploration**. To focus search results, an array of search filters can be applied. Users can decide to search for disease trajectories that occur in the entire population or one of the two sexes, only. This can be selected in the side-panel drop-down menu, when the Information tab is selected. A more detailed search filtering is further obtained by adjusting three different parameters which are trajectory length, RR and number of patients. All search filters can be adjusted independently (Fig. 5). As mentioned above, DS (ICD-10 code Q90) is a rare condition typically diagnosed at birth. Consequently, the node for Q90 only has outgoing edges and relatively few people follow each trajectory (Fig. 3). Oppositely, a search for HT (ICD-10 code I10) that is often diagnosed in older, multimorbid patients results in disease history reaching statistical significance (incoming edges) and many subsequent comorbidities (outgoing edges; Fig. 5).

Users can navigate the visualisation interface by applying tools from the toolbar to a specified part of the network or navigate manually with the curser over the visualisation interface. Tools include selecting incoming or outgoing edges to a node of choice, Neighbourhood button (selecting all incoming and outgoing edges to a node of choice) and Select inverse button (selecting anything which is not selected at that moment). Users can also move and delete nodes as well as edges manually by clicking and dragging in the interface. Nodes and edges can be deleted either individually or grouped (Figs. 5 and 6). Multiple selection is

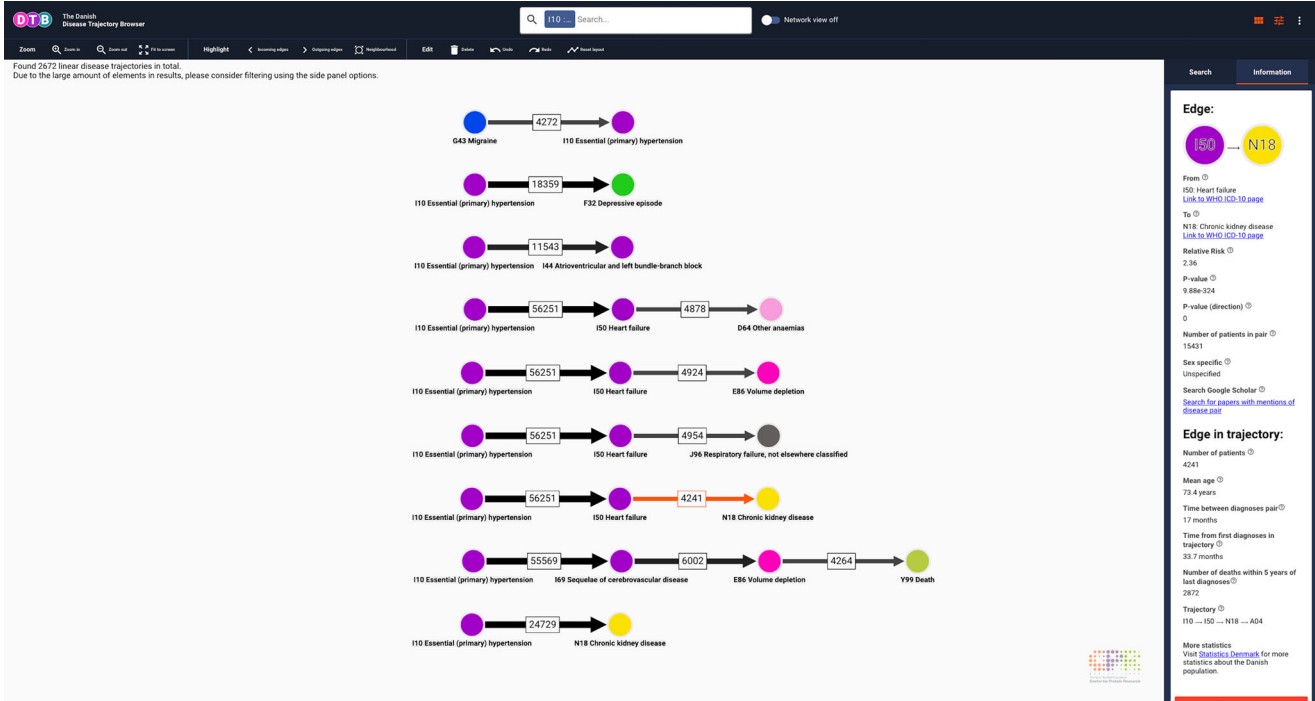

**Fig. 4 Linear disease trajectories for essential (primary) hypertension (I10).** When an edge is selected, users can explore characteristics for the entire trajectory and the individual trajectory edges separately. In this case, there are 56,251 patients who follow the trajectory I10 -> I50, 24,729 patients that follow the trajectory I10 -> N18 and 4241 patients who follow the trajectory I10 -> I50 -> N18 (visualisation interface). In the Information tab users can find information on the selected length 2 trajectory as well as the entire trajectory (side panel). If users select a different edge or search for I50 or N18, the browser allows to compare e.g. mean age at assignment of first trajectory across all statistically significant trajectories with RR > 1 in the entire population. Search filter settings: all patients, length 4–5, RR 1.15–361, number of patients 4000–114,099. I10: ICD-10 code for essential (primary) hypertension. I50: ICD-10 code for heart failure. N18: ICD-10 code for chronic kidney disease. ICD-10 International Classification of Diseases, 10th Revision. Colours of nodes represent ICD-10 chapters.

obtained by dragging with the mouse cursor while holding down the control-key (Windows/Linux) or command-key (Mac). Undo and redo buttons are available in the toolbar for simple corrections. Users can zoom in and out; either via the toolbar zoom buttons or directly by scrolling with the mouse wheel or trackpad.

**Disease trajectory analysis and data export**. In addition to interactive data analyses and visualisations, DTB allows data exports and downloads via options in the side panel, Information tab (Fig. 3). If the Search tab is selected in the side panel, options for annotation of disease nodes and transition edges are available via the drop-down menus. Node annotation offers four options (ICD-10 code and description, ICD-10 code, Diagnosis code description and None), and Edge annotation offers three options (Number of patients, Relative risk, and No annotation; Fig. 5). Nodes remain constant irrespective of selection, whereas the edge will change thickness and shade of grey once an annotation is selected. The higher the annotation value (number of patients or relative risk) the thicker and darker the edge will be. Further exploration of disease trajectories is achieved by clicking on each transition edge in the network and selecting the Information tab in the browser side panel. The user will find the relative risk, P-values, number of patients, time between diagnoses, mean age at the trajectory inception, and 5-year mortality count at trajectory end. Moreover, users can also search for diagnosis pairs at Google Scholar via a direct link and choose to delete edges or nodes (Fig. 3).

To create an overview of the raw summary statistics data, data can be summarised in a tabular format by clicking View data table

in the side panel in the Information tab. All trajectories made with DTB can be downloaded in several formats for further data processing, analyses, and presentation of search results. Cytoscape JSON format exports are available for downloading[43]. The Cytoscape files can be opened locally in the Cytoscape desktop application for rearranging or modifying the visualisation further. Export of search results as linear disease trajectories and disease trajectory networks is also possible in PNG format (with transparent background) or JPEG with white background. The raw comma-separated data values (CSV) are available for download for easy data import into Excel or data science programming applications.

## Discussion

In a modern healthcare system, the value of examining disease associations at the population-level, in both chronic conditions (exemplified above) and acute, clinical conditions can only be underscored[44,45]. Particularly, the development of computational methods for harmonising, analysing and visualising large, heterogeneous electronic health data sets is an integral part of modern medicine that is increasingly being shaped by precision medicine initiatives[19,46–48]. To date the only large disease cohort visualisation tool made publicly available is the Comorbidity-viewer from Stockholm Electronic Patient Records (SEPR) originating from ~605,000 patients[49]. However, the Comorbidity-viewer does only report disease co-occurrences, not progression patterns, and no disease trajectories or disease networks can be created. Researchers who have their own large patient registry cohorts can use existing software tools for computing comorbidity scores and analysing their own data. In contrast, DTB

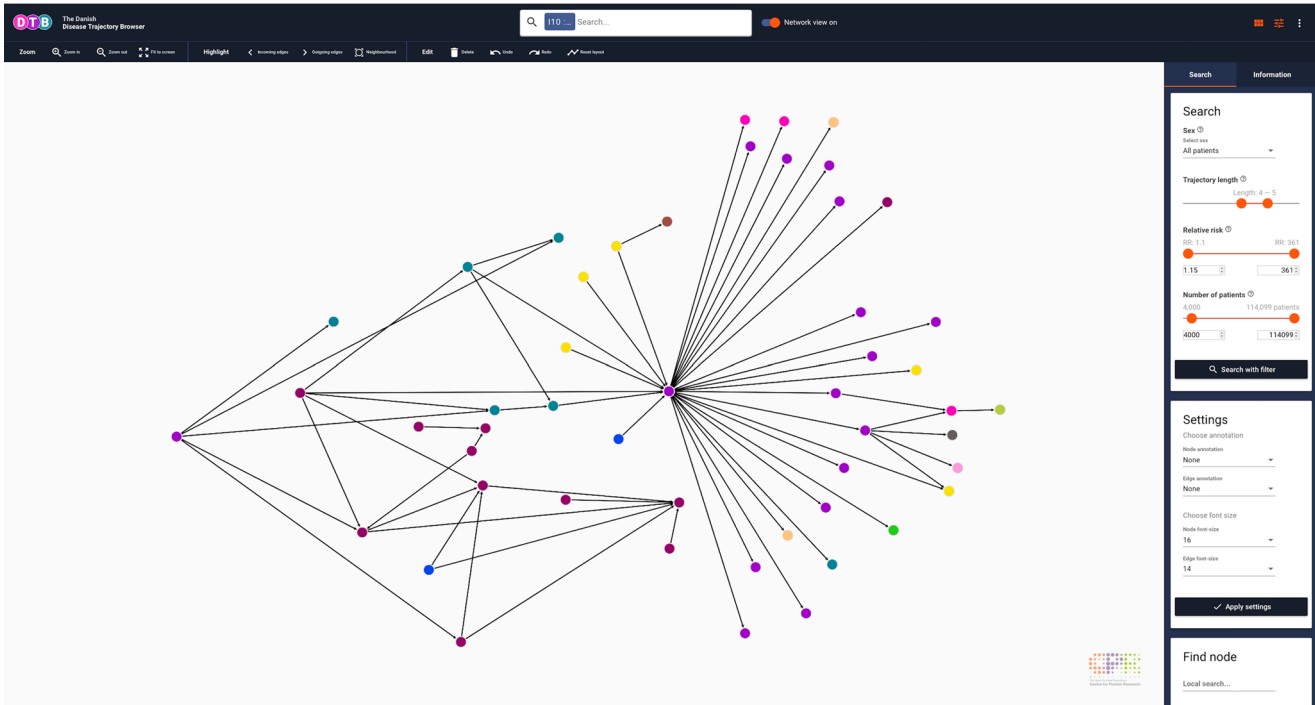

**Fig. 5 Disease trajectory network for essential (primary) hypertension (I10).** When the search tab is selected in the side panel, users can perform advanced DTB queries by application of DTB functionality, comprised of a the drop-down menu (All patients, female patients only or male patients only), notches (trajectory length, relative risk and number of patients) or fields for typing the boundary values (relative risk and number of patients). Users can customise the layout of the search results by selecting annotation (type and font) and perform a local search by typing an ICD-10 code or a disease (bottom, left corner). Search filter setting: all patients, length 4–5, RR 1.15–361, number of patients 4000–114,099. DTB Danish Disease Trajectory Browser, ICD-10 International Classification of Diseases, 10th Revision. Colours of nodes represent ICD-10 chapters.

provides useful clinical, temporal comorbidity analyses without requiring user-provided clinical data.

DTB aims to deliver easy access to a comprehensive population-wide data set of disease trajectories and comorbidities and is intended as a framework for studying disease progression patterns, where analyses with additional datatypes (e.g. socio-economic and molecular level data) can be incorporated. The browser can help researchers obtain an overview of the disease progression in a homogeneous, mostly Northern European population. Therefore, search results for diseases with a relatively low or high prevalence in Scandinavia will not necessarily replicate in other populations. Malaria represents one such example in effect of an endemic nature. Malaria is indeed included in the NPR registry, however with low incidence in Denmark. Our cut-offs have excluded malaria from the list of significant directional diagnosis pairs but DTB offers the possibility of inspecting the full list of pairs where malaria is included. The overview is summarised in statistical metrics, such as mean age at which the different trajectories started for the patients, the overall relative risk of getting one disease after the other and the 5-year mortality count for patients following specific trajectories. Such data can then be compared to statistics in other populations with akin gross domestic product (GDP), health care model, and geography or be used to characterise the degree of similarity between countries with different demographics[50]. Importantly, search results from DTB can complement existing epidemiological methods by identifying sequential trends in disease progression patterns of up to six diseases.

It is evident, that diseases tend to display heterogeneity and manifest in different phenotypic contexts[1]. The results from this study provide a framework for studying cases where the diagnosis pairs D1 -> D2 and D2 -> D3 are both statistically significant in terms of direction, even if quite few patients will follow both

pairs, but still share D2. A case like this highlights the heterogeneity within disease conditions, and points at D2 and its involvement in different disease courses. The browser functionality will allow users to identify such cases as counts are displayed for how many patients will follow D1 -> D2 -> D3, but also how many of the patients following D1 -> D2 would not follow D2 -> D3, in a longer trajectory (Fig. 4). This makes it possible to discover cases of intrinsic heterogeneity of diseases.

Many more generalised systems biology tools exist for visualising networks of biological data[51,52]; however, these tools have a molecular focus and are not suited for large-scale disease trajectory analysis[1]. Eventually, our tool can motivate data integration at the molecular and disease systems biology levels enabling researchers to further understand the underlying relationships between diseases[53]. Researchers can use the browser to compare or verify trends in their own settings, make new discoveries and hypotheses on factors driving diseases, comorbidities, and their functional relationships.

The browser's disease database is coded in the ICD-10 system which is used by WHO and many healthcare systems world-wide. It therefore facilitates entry into the system for clinicians. In a clinical setting, disease trajectories may hint at a specific treatment strategy when a doctor cares for a patient, prescribes medicine or informs patients of possible future complications and diseases. DTB will allow anyone to do disease trajectory analyses in a real-world setting, without direct access to person-sensitive registry data and without requiring access to high performance computers.

## Methods

**Underlying comprehensive population-wide registry data.** NPR contains data from all encounters at Danish hospitals, that be inpatient wards, outpatient clinics, and emergency room visits. As it has been mandatory for hospitals to report any

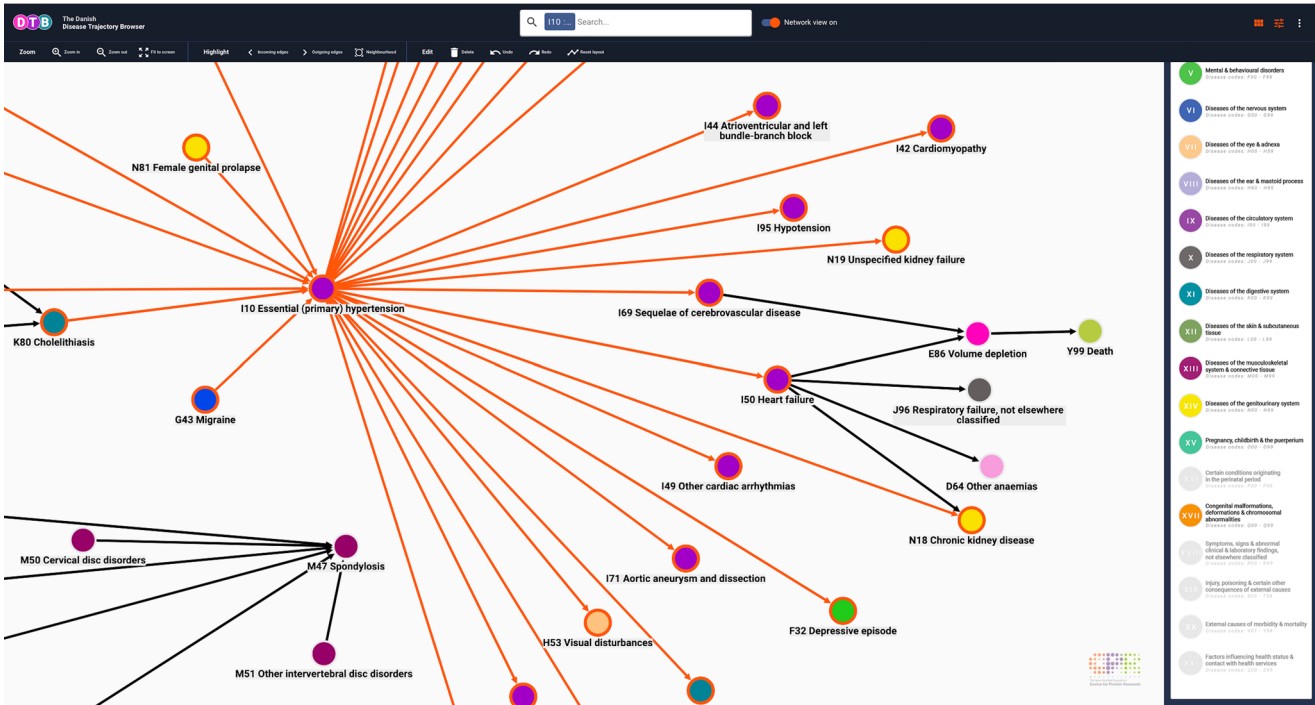

**Fig. 6 Details relating to Fig. 5 where the Neighbourhood tool has been selected.** When network view is on, users can customise the network display by applying different tools, e.g. Neighbourhood where all edges that link directly to a node are highlighted (red outline). In this case, I10 was selected and highlighted edges link to neighbour nodes. Other tools include Incoming edges, Outcoming edges and Select inverse. If the Select inverse functionality is activated after Neighbourhood, nodes and edges which are not neighbours will be highlighted. This is especially useful for deleting a section of the network, that users do not wish to export. Users can export the network that is displayed in the visualisation interface and may select parts manually by clicking on nodes and edges while holding down shift or dragging while holding down control (Windows/Linux) or command (Mac). I10 ICD-10 code for essential (primary) hypertension, ICD-10 International Classification of Diseases, 10th Revision. Colours of nodes represent ICD-10 chapters.

type of contact (inpatient, outpatient, and emergency room visits) to the NPR for reimbursement purposes since 1994, the register is complete in nature and does not suffer from population bias[27], i.e., bias in terms of demography and areas of specialty is limited. NPR also holds data on place of residence (municipality), a patients' sex, hospital and ward, hospitalisation duration, date of death, etc.[26,27]. The registry thus contains information at the level of individual patients that in relation to accuracy has been benchmarked extensively[27,54]. For disease classification, NPR uses the WHO scheme ICD-10. The ICD-10 system has been used in Denmark since 1994, where it replaced the ICD-8 system (used in Denmark from 1977 to 1994). To incorporate death in the disease trajectory analysis we added the code, Y99, defined as death within 5 years of assignment of diagnosis D1 in the trajectory D1 -> Y99. Data from The Danish Register of Causes of Death registry was used to compute the pairs needed to include mortality as a functionality in the browser[28]. The temporal disease trajectories were then calculated from this data foundation following a two-step process.

**Ethical approval**. The study was approved by the Danish Data Protection Agency (ref: 2015-54-0939 and SUND-2017-57) and Danish Health Authority (ref: FSEID-00001627 and FSEID-00003092).

**First step of the disease trajectory programme**. The first step operates in a sequential manner by first identifying statistically significant diagnosis pairs (i.e. diseases that co-occur temporarily) and then selecting the diagnosis pairs with a significant directionality, which is defined as disease trajectories of length 2. The second step concatenates these pairs together to build disease trajectories of length >2 (Fig. 1). Computing all 3,155,952 possible disease pair permutations is too computationally demanding. Therefore, a pre-filtering step is applied. The pre-filtering step estimates the $P$-values for all possible diagnosis pairs within a 5-year time-window exploiting a single Bernoulli trial, that considers each sampling of a comparison discharge a Bernoulli trial. Using the probability distributions from these Bernoulli trials, 212,008 diagnosis pairs passed the pre-filtering step at level of significance $P < 1.21 \times 10^{-9}$ (Bonferroni corrected for multiple testing) ensuring model robustness[3] (Fig. 1). For the diagnosis pairs (i.e. comprising diagnoses D1 and D2) that pass the pre-filtering step, patients are assigned diagnosis D1 are matched with $N = 10,000$ randomly selected individuals controlling for sex, age, discharge type and discharge week to limit any bias regarding seasonal disease trends. The RR is calculated as the ratio of D2 occurrences between the exposed population ($C_{exposed}$), patients who have been assigned diagnosis D1, and control

populations ($C_1 \ldots C_N$) 10,000 random controls matched to each patient with diagnosis D1.

$$\mathrm{RR} = \frac{C_{\mathrm{exposed}}}{\frac{1}{N}\sum_i C_i} \qquad (1)$$

$P$-values are estimated using a binominal distribution that models each single comparison of a disease pair with the cut-off threshold of $1.21 \times 10^{-8}$ (Bonferroni corrected for multiple testing). For the diagnosis pairs that are statistically significant the directionality is computed. This procedure identifies the pairs, D1 -> D2, where D1 occurs significantly more before D2 compared to the opposite direction using a binomial distribution. Only the diagnosis pairs with a significant directionality are retained for visualisation purposes ($P$-value < 0.05), whereas all significant diagnosis pairs with RR < 1 can be downloaded from the browser (Fig. 2 and Supplementary Data 1). To compute sex-specific disease trajectories, directional diagnosis pairs are also computed separately for male and female populations. Results from this analysis appears as a DTB functionality in the form of a search filter (Fig. 5).

**Second step of the disease trajectory programme**. In the second step, the algorithm builds the linear trajectories by concatenating statistically significant directional diagnosis pairs. For example, if D1 -> D2 and D2 -> D3 are significant directional diagnosis pairs, the pairs are fused into a disease trajectory of length three, i.e. D1 -> D2 -> D3. This principle can be re-applied to obtain longer trajectories. If the disease pair D3 -> D4 also exists, the algorithm generates the trajectory D1 -> D2 -> D3 -> D4. Patients follow the trajectories if the first occurrence of each of the diseases follows the specified trajectory. For example, one patient could be diagnosed with D1 -> X -> D2 -> Y -> D3 -> Z and thereby follow the trajectory as all three diseases (D1, D2 and D3) are present in the specified order. A trajectory is included if a minimum of 20 patients follow D1 -> D2 -> D3 without skipping any of the diseases. This can be adjusted by application of the corresponding search filter.

**Database creation**. The DTB web application contains the resource of summarised trajectory data and is queried on the fly for making linear disease trajectories and disease trajectory networks. As of now, the database content is limited for use within the web application, but portions of the database can be exported for use locally in several different formats. Formats include CSV format, JSON either for importing visualisations into Cytoscape Desktop[43] or for use in scripts or

images files with white or transparent background (i.e. PNG or JPEG formats). Supplementary Data not used in the browser, with diagnosis pairs where RR < 1 can be downloaded from the side panel Information tab (Fig. 2).

The tool was developed with the use of the graph theory Javascript library called Cytoscape.js and implemented using the web development framework Angular v6 by Google, running on a Node.js backend server (see below).

DTB has information on all ICD-10 chapters but Certain conditions originating in the perinatal period (chapter XVI), Symptoms, signs & abnormal clinical & laboratory findings (chapter XVIII), Injury, poisoning & certain other consequences of external causes (chapter XIX), External causes of morbidity and mortality (chapter XX), and Factors influencing health status & contact with health services (chapter XXI). In the Information tab in the browser side panel, excluded chapters are grey (Fig. 6). The tool is released as a web application on the URL: http://dtb.cpr.ku.dk and gives access to a an interactive interface for exploring, analysing and visualising the Danish National Patient Register cohort of 7.2 million patients recorded over almost 25 years.

**Software: frameworks and libraries used**. Cytoscape.js – graph frontend Javascript library.

Dagre.js – directed graph Javascript library.
Angular (frontend Javascript framework).
Node.js (Sails.js) – backend Javascript framework.
MongoDB (database software).

## Data availability

Permission to access and analyse the underlying person-sensitive data can be obtained following approval from the Danish Data Protection Agency and the Danish Health Authority. A statistical summary for this article is available as a Supplementary Data 1. Due to privacy concerns, the browser and the provided Supplementary Data 1 only contain diagnosis and co-occurrence information when it has been assigned to at least five patients. All data made available are non-person-sensitive summary level data.

## Code availability

To analyse the data the following software tools were used: R v. 3.4.0, Python v. 2.7, C++ v. 11 and Cytoscape Desktop v. 3.6.0. The key algorithm has been described in published literature (see refs. [3,14]).

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

## Acknowledgements
We would like to acknowledge funding from the Novo Nordisk Foundation (grant agreements NNF14CC0001 and NNF17OC0027594).

## Author contributions
T.S. and S.B. conceived the study. S.B. obtained the funding. T.S., R.R., I.F.J., A.D.H. and S.B. performed the literature search, figures, study design, programming and data analysis. T.S., R.R., I.F.J., A.D.H., M.L., A.A-O., J.X.H., A.B.J., K.B. and S.B. contributed to data interpretation. T.S., R.R., I.F.J., A.D.H. and S.B. wrote the initial draft, and M.L., A.A.-O., J.X.H., A.B.J., K.B. and S.B. contributed also to the final article.

## Competing interests
S.B. reports ownerships in Intomics A/S, Hoba Therapeutics Aps, Novo Nordisk A/S, Lundbeck A/S and managing board memberships in Proscion A/S and Intomics A/S outside the submitted work.
