## [Peer Review File · Nature Communications]

Reviewers' Comments:

Reviewer #1:

Remarks to the Author:

The authors describe a disease trajectory browser for exploring disease progression patterns within longitudinal Danish patient data consisting of the entire Danish population. They use two examples to demonstrate its use - patients with Down syndrome, and those with primary hypertension.

The browser currently seem to be solely based on ICD10 diagnostic codes. The main issue I have is that it is difficult to see how this tool will be used, other than for simple descriptions of direct disease relationships, given that it does not seem to allow for adjustment or exploration of factors such as demographics (sex, ethnicity, etc) that are known to have important associations with disease and disease trajectories; neither can potential mechanisms e.g. lifestyle factors be explored. This appears to be a significant limitation. Furthermore, I have tested the browser (admittedly, as a novice user) using for example the ICD10 codes for mental retardation/ intellectual disability - a more common condition than Down syndrome (although nearly all individuals with Down syndrome may have intellectual impairment and thus a diagnosis of mental retardation). Intellectual disability is known to be associated with numerous comorbid conditions over time, but the search in the browser resulted in very few associations - limited to 6 incoming edges, and no outgoing ones. Down syndrome was not one of the incoming edges.

From that point of view, existing epidemiological approaches and analysis methods of longitudinal datasets appear to have more value in understanding disease trajectories than this browser. I would suggest the authors consider in more detail potential uses of the browser, and contrast those to existing epidemiological methodology.

In addition, the authors should describe if ongoing development of the browser is being considered.

Reviewer #2:

Remarks to the Author:

This is a very interesting and innovative paper which adds considerable value to the underlying Danish national data. The associated web application which allows exploration of disease trajectories is easy to use and accessible.

To a large extent the paper serves as a "shop window" onto the associated web application, and that in itself is valuable. However, there are some unavoidable limitations that would benefit from further discussion in the paper. One issue is that the scope of the system is necessarily limited by the range of morbidity experienced by the Danish population - so for example looking for malaria doesn't yield any results. Whilst this kind of limitation is inevitable, putting the system into the international arena means that the Danish context of the data needs to be more clearly discussed.

The other major issue that is not addressed - and maybe it can't be from the underlying data - is any identification of individual morbid episodes as specifically resulting in death. In other words, being able to pull out the disease trajectories leading to deaths due to specific causes would add considerable value to the whole system. If that is impossible, the authors should say so; if it is possible but not yet done, it should be mentioned as a future agenda.

Reviewer #3:

Remarks to the Author:

This manuscript describes an innovative and ambitious project to characterize the progression of illness from one disease to development of another disease. The project is linked to an intriguing online tool that allows for graphing the network of connections among diseases. The online tool is generally nice and flexible. There were a few clunky pieces from my browser (Chrome) so it would probably benefit from a little more user feedback but nothing was so serious it would have stopped me if I had a question that could have been answered with the data. There are some important missing pieces and lost opportunities, which I hope the authors will consider addressing:

- The disease trajectory browser (DTB) restricts to disease pairs for which there is an elevated risk,

omitting all for which there was a reduced risk. This throws away half the information in the original data. The fact that one condition predicts much lower probability of another is quite informative and may offer novel insights into disease etiology. This is a simple modification of the framework such that $RR < 1$ is not omitted.

- Similarly, showing diseases that co-occur temporally would be useful. There is no clear reason to omit diseases for which occurrence tends to be simultaneous (which is what I understand the current protocol does). Just show that these are co-occurring in time.

- The disease pathways are built up pairwise, so if $D1 \rightarrow D2$ and $D2 \rightarrow D3$, it is assumed that $D1 \rightarrow D2 \rightarrow D3$. This may not always be the case, however, and it would be incredibly useful to test this. In cases in which $D1 \rightarrow D2$ and $D2 \rightarrow D3$ but $D1$ does not predict $D3$, it suggests important heterogeneity in $D2$ – ie. The type of $D2$ that follows $D1$ is not the same as the type of $D2$ that predicts $D3$. The major insights from personalized medicine have derived from recognizing heterogeneity within disease conditions, and this is an application well-designed to do so. Assuming that the pairwise correlations translate into sequences is an unfounded assumption and could lead to important misinterpretations. This would be easy to incorporate into the existing framework by testing not just $D1 \rightarrow D2$ and $D2 \rightarrow D3$ but also $D1 \rightarrow D3$ and even $D1$ predicting $D3$ conditional on $D2$ (side note: perhaps too ambitious for the current paper but describing the conditional associations would open up a whole new field of questions).

- It should be clarified in the Results that your matching procedure accomplishes age, sex, and period adjustment.

- Do I understand correctly that the matching procedure matches to other patients, not to the population as a whole? This does not seem the correct calculation and it is not necessary since the coverage of the Danish population is nearly complete. The comparison should be to the rest of the population without that diagnosis in the week of the index individual's diagnosis, not the rest of the population with an equivalent discharge type in that week. This harkens back to the days when epidemiology was filled with hospital based case control studies and spurious associations were induced by the nature of the control group selected. Please also detail the discharge type categories.

- With respect to step 1 of identifying disease pairs, is the single binomial test age adjusted and is it "ever diagnosed" or somehow taking time-at-risk into account? I believe if this first step is not correctly handled you risk building certain biases into your findings. I do not see the justification for step 1. Why not start with the matched study in step 2?

- Please clarify in step 2 how the relative risk is calculated. Does it account for time at risk? Or is it truly relative 'risk of ever being diagnosed'? If the latter, this will create associations between diagnoses that have a long life expectancy and diagnoses that occur in late life. In general, there needs to be much more information about time at risk and a comparison of incidence rates, not (cumulative) risk ratios.

- It would be incredibly valuable to include the death rate for people on each trajectory.

- Small point but the number of decimal places in figure 1 is distracting.

- A flow chart of how many disease comparisons were evaluated and excluded versus retained to develop the final database would be helpful.

Response to referee comments.

Reviewer #1 (Remarks to the Author):

The authors describe a disease trajectory browser for exploring disease progression patterns within longitudinal Danish patient data consisting of the entire Danish population. They use two examples to demonstrate its use - patients with Down syndrome, and those with primary hypertension.

The browser currently seem to be solely based on ICD10 diagnostic codes. The main issue I have is that it is difficult to see how this tool will be used, other than for simple descriptions of direct disease relationships, given that it does not seem to allow for adjustment or exploration of factors such as demographics (sex, ethnicity, etc) that are known to have important associations with disease and disease trajectories; neither can potential mechanisms e.g. lifestyle factors be explored. This appears to be a significant limitation.

Response: We have now added functionality that enables users to search for sex-specific trajectories (males or females, see modified figure 5), in addition to the full population as in the submitted version. For an in-depth characterization of population-wide differences in disease progression patterns in men and women we refer to our Nature Communication paper from 2019 (ref. 14). Additionally, we have added 5-year mortality for all trajectories (Methods and Results, see modified figure 3). We have no information on ethnicity, lifestyle data or socioeconomic data in the patient registry. Integrating socioeconomic data and disease trajectories is a major task that is not in the scope of this paper. See also the response to the further development of the browser below.

Furthermore, I have tested the browser (admittedly, as a novice user) using for example the ICD-10 codes for mental retardation/intellectual disability - a more common condition than Down syndrome (although nearly all individuals with Down syndrome may have intellectual impairment and thus a diagnosis of mental retardation). Intellectual disability is known to be associated with numerous comorbid conditions over time, but the search in the browser resulted in very few associations - limited to 6 incoming edges, and no outgoing ones. Down syndrome was not one of the incoming edges.

Response: Importantly, the browser addresses directional disease co-occurrences and not co-occurrences in general. Down syndrome and intellectual impairment/mental retardation is not an ideal positive control, as intellectual impairment is a potential characteristic of Down syndrome and not a comorbidity per se. However, the test case that the referee is presenting illustrates important prerequisites when working with ICD-10 annotated diseases. Intellectual disability can be coded as several different ICD-10 codes: Mental retardation (F70-F79). The search results will depend on the code the user searches for. Thus, this case reflects that the co-morbidity patterns, vary with the disease severity and specificity of the code (varying from mild mental retardation, F70 to profound mental retardation, F73 and unspecified, F79), as one would also expect. The best way of searching for more general conditions like intellectual disability might be to include all relevant codes in the search to get the full overview. Alternatively, users may also search for the comorbidities in the underlying condition. In this case unspecified mental retardation is included in the search for Down

syndrome (figure 3). Also, diseases are only included if they are primary or secondary hospital diagnoses (inpatient/outpatient/emergency), thus some types of intellectual disability may be underreported.

From that point of view, existing epidemiological approaches and analysis methods of longitudinal datasets appear to have more value in understanding disease trajectories than this browser. I would suggest the authors consider in more detail potential uses of the browser, and contrast those to existing epidemiological methodology.

Response: The value of the browser compared to existing epidemiological approaches is that it operates in a disease spectrum-wide mode, independent of an index-disease, while existing epidemiological approaches often are very specific (e.g. selected diseases in a hypothesis-driven manner). Further, no epidemiological methods account for the order of up to five diseases, but adjust normally for the presence of co-morbidities (i.e. covariates). As we state in the Discussion, we see traditional approaches and the data-driven disease trajectory model as complementary, and not in conflict.

In addition, the authors should describe if ongoing development of the browser is being considered.

Response: In the Discussion, we have now added comments on the planned development of the browser as a framework for studying population-wide disease progression patterns. Eventually the browser may also include other datatypes. In fact, we have a large grant that deals with integrating socioeconomic data and disease trajectories and we intend to add functionalities to the browser from the results of this six-year project. The Danish socioeconomic data are also population-wide, but it is a major task to add data on employment, social benefits, family size, etc. etc. which we find is beyond the scope of this paper. In the grant we collaborate with the Statistics Denmark (DST) organization, and have added a link in the browser to the DST (e.g. figure 3). Their web pages contain information on the data they hold, e.g. quality of life metrics, social conditions and education.

Reviewer #2 (Remarks to the Author):

This is a very interesting and innovative paper which adds considerable value to the underlying Danish national data. The associated web application which allows exploration of disease trajectories is easy to use and accessible.

Response: We thank the reviewer acknowledging the innovative aspect of our work.

To a large extent the paper serves as a "shop window" onto the associated web application, and that in itself is valuable. However, there are some unavoidable limitations that would benefit from further discussion in the paper. One issue is that the scope of the system is necessarily limited by the range of morbidity experienced by the Danish population - so for example looking for malaria doesn't yield any results. Whilst this kind of limitation is inevitable, putting the system into the international arena means that the Danish context of the data needs to be more clearly discussed.

Response: We entirely agree with the referee and have now discussed this limitation further, including the fact that disease trajectories not necessarily will replicate in all populations

irrespective of GDP, health care model and geography. We think it is important to be able to compare across countries and in that way for example identify cases where diseases have genetic etiologies versus those which are related to other, modifiable exposures (multifactorial diseases). Malaria is indeed included in the national registry, however as the reviewer says with low incidence in Denmark. We have now added a possibility for inspecting the full list of significant disease pairs (Supplementary Information) and have elaborated further on this in the Discussion. Users can now inspect the list of all statistically significant pairs by downloading the ones with $RR < 1$ (e.g. malaria), which is also exemplified in the paper (figure 2).

The other major issue that is not addressed - and maybe it can't be from the underlying data - is any identification of individual morbid episodes as specifically resulting in death. In other words, being able to pull out the disease trajectories leading to deaths due to specific causes would add considerable value to the whole system. If that is impossible, the authors should say so; if it is possible but not yet done, it should be mentioned as a future agenda.

Response: This was already on our list of potential future improvements. We have now included death as an event (Y99) and have used the Danish Register of Causes of Death to compute the pair information needed to include this feature (see Methods, ref. 28). We have also included information on “death ratio” as 5-year mortality after a trajectory in the browser (see further comments under reviewer #3 who has a similar suggestion). This means that users can now search for all disease trajectories that include Y99 by submitting it as a query and within a search for e.g. hypertension, identify the disease trajectories that contain Y99 (side bar, Information tab, “Find node” functionality). Users can find the 5-year mortality of disease trajectories resulting from a search query in the Information tab (figure 3).

Reviewer #3 (Remarks to the Author):

This manuscript describes an innovative and ambitious project to characterize the progression of illness from one disease to development of another disease. The project is linked to an intriguing online tool that allows for graphing the network of connections among diseases. The online tool is generally nice and flexible.

Response: Thank you for acknowledging the level of ambition behind the method and the tool. There were a few clunky pieces from my browser (Chrome) so it would probably benefit from a little more user feedback but nothing was so serious it would have stopped me if I had a question that could have been answered with the data. There are some important missing pieces and lost opportunities, which I hope the authors will consider addressing:

Response: We have now added a note in the manuscript for time-optimizing searches and added a message in the browser to users that prepares for potential wait time.

- The disease trajectory browser (DTB) restricts to disease pairs for which there is an elevated risk, omitting all for which there was a reduced risk. This throws away half the information in the original data. The fact that one condition predicts much lower probability of another is quite informative and may offer novel insights into disease etiology. This is a simple modification of the framework such that $RR < 1$ is not omitted.

Response: As mentioned above (under reviewer #2) we have now included a list of all pairs and this list also includes relative risks which are below 1.0 (Supplementary Information). We agree that inverse comorbidities represent a very active and hot research area where several papers emerged recently (e.g. Catalá-Lépez et al, *Psychother Psychosom*, 2014). However, our tool is meant for disease progression analysis and not for discovering inverse comorbidities per se. Inverse comorbidity statistics is very sensitive to underreporting and to the discrimination between hospital diagnoses, treatment or exposure information to be found elsewhere, such as smoking. It is therefore not as simple as stated by the reviewer that we throw half of the information away because one needs to correct for these other issues in quite complicated ways. However, we have now added the information the reviewer asks for at the level of population-wide disease pairs for users to download and thus provide the missing half the reviewer mentions (e.g. figure 2). To build this information into an automatized disease trajectory tool for inverse co-morbidities is outside the scope of the current paper. Papers on inverse comorbidities are in fact highly specific and aims for carefully controlling for or excluding some of the factors mentioned above.

- Similarly, showing diseases that co-occur temporally would be useful. There is no clear reason to omit diseases for which occurrence tends to be simultaneous (which is what I understand the current protocol does). Just show that these are co-occurring in time.

Response: As mentioned above we now make more visible how to find the list in the DTB of all diseases that co-occur with an $RR < 1$ (figure 2), and provide the full list of all disease pairs with the manuscript (Supplementary Information). Moreover, we have explained the disease trajectory method in greater detail in Methods and provided a flow-chart (figure 1). From the Supplementary Information it is possible to inspect the disease co-occurrences with $RR < 1$. The table in the Supplementary Information with all disease pairs is of considerable size (77,294 entries) and downloadable as a json-file.

- The disease pathways are built up pairwise, so if $D1 \rightarrow D2$ and $D2 \rightarrow D3$, it is assumed that $D1 \rightarrow D2 \rightarrow D3$. This may not always be the case, however, and it would be incredibly useful to test this. In cases in which $D1 \rightarrow D2$ and $D2 \rightarrow D3$ but $D1$ does not predict $D3$, it suggests important heterogeneity in $D2$ – ie. The type of $D2$ that follows $D1$ is not the same as the type of $D2$ that predicts $D3$. The major insights from personalized medicine have derived from recognizing heterogeneity within disease conditions, and this is an application well-designed to do so. Assuming that the pairwise correlations translate into sequences is an unfounded assumption and could lead to important misinterpretations. This would be easy to incorporate into the existing framework by testing not just $D1 \rightarrow D2$ and $D2 \rightarrow D3$ but also $D1 \rightarrow D3$ and even $D1$ predicting $D3$ conditional on $D2$ (side note: perhaps too ambitious for the current paper but describing the conditional associations would open up a whole new field of questions).

Response: We agree that this was not described in a sufficiently clear manner. We only display trajectories of length ≥ 2 , if a certain number of patients follow the entire trajectory. This was for example also discussed in another paper from our group dealing with analysis of sepsis trajectories (ref. 21) and also in a paper on pre-cancer development (ref. 23). The suggestion of also testing $D1 \rightarrow D3$ is already done, in our approach, as all pairs are tested and the results have now been added as supplementary material (Supplementary Information). The reviewer highlights the heterogeneity within disease conditions, and points at $D2$ and its involvement in different disease courses. We thank the reviewer for formulating this so clearly. The numbers the reviewer asks for (e.g. how many

patients following D1->D2 would *not* follow D2->D3) could actually be found already, but with difficulty. We have now added this information in a more accessible way in the browser window, that is, when clicking on a specific pair, Dx->Dy, how many patients, *in a longer trajectory*, would *not* follow this particular pair (figure 4). This functionality makes it possible to discover cases of intrinsic heterogeneity of diseases that the reviewer highlights. We have now added to the Discussion along the lines of what the reviewer requests and also in Results.

- It should be clarified in the Results that your matching procedure accomplishes age, sex, and period adjustment.

Response: We agree with the reviewer and have made this clearer, but also mentioned the more elaborated discussion of the matching procedure in our Nature Communications 2014 paper (ref. 3). The matching procedure has now been clarified Results. On that same note we have now divided the previous section: Base pair and trajectory computation in two to further call attention to the steps in the disease trajectory algorithm. As mentioned above we have now also included the breakdown on both sexes per the comment from reviewer 1 and clarified it in Results.

- Do I understand correctly that the matching procedure matches to other patients, not to the population as a whole? This does not seem the correct calculation and it is not necessary since the coverage of the Danish population is nearly complete. The comparison should be to the rest of the population without that diagnosis in the week of the index individual's diagnosis, not the rest of the population with an equivalent discharge type in that week. This harkens back to the days when epidemiology was filled with hospital-based case control studies and spurious associations were induced by the nature of the control group selected. Please also detail the discharge type categories.

Response: It is correct that the coverage of the Danish population in NPR is nearly complete, and hence the method compares to the Danish population as a whole. However, for each case patient, the control group consists of 10,000 individuals and it is not selected in any biased way (cf. randomly, Methods). To use the entire population of 7M patients as control group would require exorbitant computational resources and the estimates are already highly conservative in terms of case/controls. Moreover, we cannot follow reviewer fully here as a non-age matched control population would lead to many spurious findings, for example when comparing children and adults (e.g. childhood asthma vs. adult asthma or children with epilepsy vs. adults where it is more often sequelae to other diseases neurological conditions). The discharge categories are described in the 2014 paper (ref. 3) and we feel that is sufficient.

- With respect to step 1 of identifying disease pairs, is the single binomial test age adjusted and is it "ever diagnosed" or somehow taking time-at-risk into account? I believe if this first step is not correctly handled you risk building certain biases into your findings. I do not see the justification for step 1. Why not start with the matched study in step 2?

Response: As the data in this model are left-censored, we cannot be certain that patients in the registry were free of all other codes before they are assigned an ICD-10 code, hence time-at-risk is not within the scope of this method and the underlying data. In principle, the time distribution between two diseases could be bimodal or even more complex and we do not address these differences but limit ourselves to address the order. We also think the reviewer misinterpreted step 1 and step 2. Step 1 identifies all significant disease pairs by calculating p-values and relative risks. Step 2 concatenates the significant disease pairs into disease trajectories. We have clarified this in

Results as well as in Methods and included a new figure intended to provide a better overview of the steps in the method (figure 1A). Justification for pre-filtering is that there are 3,155,952 possible disease pairs and testing them all would be too computationally demanding thus the prefiltering step is needed (reviewer #3).

- Please clarify in step 2 how the relative risk is calculated. Does it account for time at risk? Or is it truly relative ‘risk of ever being diagnosed’? If the latter, this will create associations between diagnoses that have a long life expectancy and diagnoses that occur in late life. In general, there needs to be much more information about time at risk and a comparison of incidence rates, not (cumulative) risk ratios.

Response: As mentioned above the relative risk is calculated in the first step of the analysis. We have clarified the calculation of the relative risk in Methods and emphasized that the analysis is not a survival analysis, but a framework for discovering disease progression patterns that extrapolates from a binomial distribution. The browser has been updated so counts in NPR for all level three ICD-10 codes with more than 5 patients can be retrieved from the browser (e.g. figure 2). We do not look at the risk of ever being diagnosed, we look at the risk of being diagnosed with diagnosis D2 within 5 years after diagnosis of D1 (earliest assignment of the diagnosis code). This reduces associations between diagnoses that have a life-long expectancy and diagnoses that occur late in life and highlight diagnosis pairs that are more likely to be associated due to a common underlying etiology. As we also mention above and in the Discussion, that we are developing this framework as an approach that will complement existing epidemiological modelling strategies.

- It would be incredibly valuable to include the death rate for people on each trajectory.

Response: We have added 5-year mortality to the browser for each single trajectory (e.g. figure 3).

- Small point but the number of decimal places in figure 1 is distracting.

Response: We have changed all numbers in the browser to two decimals (e.g. figure 3).

- A flow chart of how many disease comparisons were evaluated and excluded versus retained to develop the final database would be helpful.

Response: We have created a flowchart of how many disease comparisons are evaluated in each step of the calculations (Figure 1A).

LIST OF ALL CHANGES MADE TO THE FUNCTIONALITY OF THE BROWSER:

- Implemented death (Y99) as an event in disease trajectories
- Calculated death ratio/5-year mortality for all disease trajectories and made it available
- Calculated number of females/males following all disease trajectories and made it available

- Integrated both female and male specific trajectory options in the browser and made them available
- Created a flowchart showing the number of disease pairs in each step of the analysis
- Added a downloadable table with all information on disease pairs with no direction or relative risk < 1 .
- Added incidence rates, number of females and males and 5-year mortality for all third level ICD-10 codes with more than 5 patients
- Added new functionality that makes it possible to extract additional counts of patients following the pairs in each step in the trajectories
- Fixed a bug when displaying the disease trajectory networks, so the browser now shows the correct number of patients per edge (disease trajectory).

Reviewers' Comments:

Reviewer #1:

Remarks to the Author:

The authors have addressed several of the reviewers' concerns, and added additional functionality, by including mortality data and allowing for selecting for sex. Nevertheless, the browser still seems to be at an early stage of development, and its value remains in my view somewhat uncertain. There are significant limitations, such as the lack of identifying a causal link between Down syndrome and mental retardation/ ID regardless of the ICD10 codes used (this is not simply a co-occurrence - Down syndrome (trisomy 21) is a recognised cause of ID).

Reviewer #2:

Remarks to the Author:

I am impressed by the flexibility of the authors' responses to comments from myself and other reviewers. It is important for us all to understand that this is not the kind of paper that presents a definitive, final result, but rather presents work in progress that will undoubtedly develop further, and has indeed done so during this review process. Thus I would be happy for it to progress to publication, as one way of furthering this overall line of work.

Reviewer #3:

Remarks to the Author:

Thank you for your revisions. Your argument that there is something uniquely difficult about evaluating associations for which the $RR < 1$ doesn't make sense, because all of those issues would be equally relevant to any disease pairing and lead you to overlook or overstate disease pairs with an $RR > 1$, but perhaps incorporating the additional information is too much to accomplish. It would also be helpful if there was a way to collapse ICD codes into larger categories, rather than specific codes. Again, this may be beyond the scope of the current project. As is, the browser should be quite useful.

REVIEWERS' COMMENTS:

Reviewer #1 (Remarks to the Author):

The authors have addressed several of the reviewers' concerns, and added additional functionality, by including mortality data and allowing for selecting for sex. Nevertheless, the browser still seems to be at an early stage of development, and its value remains in my view somewhat uncertain. There are significant limitations, such as the lack of identifying a causal link between Down syndrome and mental retardation/ ID regardless of the ICD10 codes used (this is not simply a co-occurrence - Down syndrome (trisomy 21) is a recognised cause of ID).

Response: We have already indicated in the manuscript that the trajectories do not prove causality. We display co-occurrences in a large nationwide database and there is no way we manually and systematically could add currently known causal correlations between diagnoses. So the limitation the reviewer points at is already commented on in the Discussion section.

Reviewer #2 (Remarks to the Author):

I am impressed by the flexibility of the authors' responses to comments from myself and other reviewers. It is important for us all to understand that this is not the kind of paper that presents a definitive, final result, but rather presents work in progress that will undoubtedly develop further, and has indeed done so during this review process. Thus I would be happy for it to progress to publication, as one way of furthering this overall line of work.

Peter Byass

Response: We thank the reviewer for acknowledging all the improvements made in the revision phase.

Reviewer #3 (Remarks to the Author):

Thank you for your revisions. Your argument that there is something uniquely difficult about evaluating associations for which the $RR < 1$ doesn't make sense, because all of those issues would be equally relevant to any disease pairing and lead you to overlook or overstate disease pairs with an $RR > 1$, but perhaps incorporating the additional information is too much to accomplish. It would also be helpful if there was a way to collapse ICD codes into larger categories, rather than specific codes. Again, this may be beyond the scope of the current project. As is, the browser should be quite useful.

Response: We would like to repeat that we do not find the disease pairs with $RR < 1$ irrelevant. We think the reviewer has overlooked that we now provide a supplementary file including all pairs with $RR < 1$. What we object to is to use these data in an automated fashion to indicate inverse comorbidities in the trajectories, this would in many cases be misleading. We respectfully disagree on the symmetry that the reviewer states, missing diagnoses represent a completely different problem compared to excess diagnoses or misdiagnoses. We will continue to work on this problem and aim for expanding the browser with inverse comorbidity functionality, but we need to benchmark this in a completely different way as no tool elsewhere provide this kind of clinical data systematically. One has to turn to molecular level data, which is a completely new endeavor.